# Maternal Reassurance, Satisfaction, and Anxiety after First-Trimester Screening for Aneuploidies: Comparison between Contingent Screening and Universal Cell-Free DNA Testing

**DOI:** 10.3390/diagnostics14111198

**Published:** 2024-06-06

**Authors:** Anna Luna Tramontano, Ilaria Marano, Giuliana Orlandi, Antonio Angelino, Maria Rivieccio, Caterina Fulgione, Giuseppe Maria Maruotti, Gabriele Saccone, Gabriella De Vita, Maurizio Guida, Laura Sarno

**Affiliations:** 1Mother and Child Department, University Hospital Federico II, 80131 Naples, Italy; tranalu@icloud.com (A.L.T.); maranoilaria@gmail.com (I.M.); 2Department of Neurosciences, Reproductive Science and Dentistry, University of Naples Federico II, 80131 Naples, Italy; giulianaorlandi2907@gmail.com (G.O.); caterina.fulgione@gmail.com (C.F.); gabriele.saccone.1990@gmail.com (G.S.);; 3Department of Public Health, University of Naples Federico II, 80131 Naples, Italygm.mar@tiscali.it (G.M.M.); 4Department of Molecular Medicine and Medical Biotecnology, University of Naples Federico II, 80131 Naples, Italy; 261092@studenti.unimore.it (M.R.); gdevita@unina.it (G.D.V.)

**Keywords:** first-trimester screening, contingent screening, aneuploidy, cell-free DNA, maternal anxiety, maternal satisfaction, maternal reassurance

## Abstract

Background: This study aims to evaluate maternal reassurance, satisfaction, and anxiety after two different strategies for the first-trimester screening for aneuploidies. Methods: Patients between 11 + 3 and 13 + 6 weeks of gestation attending the first-trimester screening at Department of Mother and Child, University Hospital Federico II, Naples, Italy have been recruited and randomly allocated to contingent screening or universal cell-free fetal DNA testing (cffDNA). Questionnaires to measure reassurance, satisfaction, and anxiety have been filled twice: (Q1) after randomization and (Q2) after receiving results. Anxiety was measured by an Italian-version short form of the state scale of the Spielberger State–Trait Anxiety Inventory (STAI); child-related anxiety was measured by the 11-item Pregnancy-Related Anxiety Questionnaire—Revised Regardless of Parity (PRAQ-R2 scale); fear of bearing a physically or mentally handicapped child was measured considering only four items (item 4, 9, 10, and 11) of the PRAQ-R2 scale. Results: 431 patients were recruited: 205 (49%) were randomized in the contingent screening arm, 226 (51%) in the cfDNA arm. Maternal reassurance, satisfaction, and anxiety were not different in the two groups. Conclusion: A contingent screening for aneuploidies in the first trimester seems able to ensure the same maternal reassurance and satisfaction as a cfDNA analysis in the low-risk population and to not affect maternal anxiety.

## 1. Introduction

Cell-free DNA testing (cfDNA) provides an effective screening for Trisomies 21, 18, and 13, which are the most common aneuploidies. According to a meta-analysis of clinical validation and implementation studies, screening by cfDNA could detect 99% of fetuses with Trisomy 21, 98% of Trisomy 18, and 99% of Trisomy 13 at a combined False Positive Rate (FPR) of 0.13% [1]. On the contrary, in a retrospective analysis of 108,892 pregnancies, a first-trimester combined screening test for aneuploidies, based on fetal ultrasound measurements, combined with a biochemical analysis of maternal serum, demonstrated a detection rate of 90%, 97%, and 92% for Trisomy 21, 18, and 13, respectively, at a FPR of 4% [2].

Even if cfDNA demonstrated a better performance than the combined screening test, its higher cost affected its implementation in clinical practice. Therefore, only few European countries offer cfDNA to all pregnant women, whereas the majority implemented it as a contingent screening after high or intermediate risk was detected at the combined screening test [3].

Several implementation studies [4,5,6,7,8,9] demonstrated that integrating cfDNA and the combined screening test in a contingent model could be a cost-effective method for the first-trimester screening for aneuploidies. However, as reported in the national guidelines [10], the available evidence on the implementation of this screening strategy does not refer to the Italian scenario. Prefumo et al. developed a health economic model to demonstrate that the use of cfDNA, as a second-line screening in the case of increased risk at the combined screening test, could improve the detection rate of aneuploidies and reduce costs for the national health system [11].

Nowadays, only two Italian Regions (Emilia Romagna and the Autonomous Province of Bolzano) implemented cfDNA in a public setting, while it is available only in private settings in all the other Regions. Despite the lack of implementation of this test in the national health system, it has been estimated that around 25–50% of Italian pregnant women use a cfDNA analysis through private clinics [3]. Its use has been largely spread among Italian pregnant women during the last few years, probably due to the reassurance deriving from the results of a test with such a good performance. Indeed, during their first trimester of pregnancy, mothers are overwhelmed by deep concerns, also due to the need to perform crucial screening tests in this period [12].

Chromosomal anomalies could determine congenital mental disability as well as structural problems; although some individuals have only mild problems and can have relatively normal lives, having a baby with aneuploidies might have a significant impact on family life [13]. The thought of a potential adverse outcome can lead to elevated levels of stress and anxiety in pregnant women.

Therefore, we think that implementation studies should also consider the psychological impact of a screening strategy on the mothers. However, so far, only few studies [14,15,16,17] have addressed this aspect.

The aim of the current study was to compare maternal reassurance, satisfaction, and anxiety in women undergoing two different screening strategies: the first-trimester contingent screening test (cfDNA given only to patients demonstrating an intermediate risk at the combined screening test) versus universal cfDNA.

## 2. Materials and Methods

### 2.1. Study Design, Setting, and Study Population

This was a single-center, open-label, parallel-group, randomized clinical trial conducted at Department of Mother and Child of University Hospital Federico II of Naples, from 1 November 2022 to 15 April 2024, as a sub-study of the primary study “A Pilot Study for Implementation of First Trimester Screening of Aneuploidies in Campania Region” (Clinicaltrials.gov NCT05798858) that is still ongoing. For the purpose of this planned, interim, preliminary analysis, we aimed to compare women’s experience after two different protocols of first-trimester screening for aneuploidies. Patients were approached during the first-trimester ultrasound screening; those who consented to participate in the study were randomized to either contingent screening or universal cfDNA testing, as summarized in Figure 1. In the contingent screening group, first, a combined screening test (first-trimester screening ultrasound + maternal biochemistry (free beta HCG and PAPPa)) was performed; second, patients were divided into three groups according to the combined screening risk, and the screening was carried on as follows: (1) low risk (<1:1000): no other test; (2) intermediate risk: (from 1:101 to 1:1000): cfDNA testing was offered; and (3) high risk (≥1:100): invasive procedure was offered. In the universal cfDNA testing group, first-trimester ultrasound screening was performed followed by cfDNA testing for the three main chromosomal abnormalities (Trisomy 21, Trisomy 13, and Trisomy 18); an invasive procedure was offered in case of positive cfDNA test.

According to the protocol of the primary study, inclusion criteria were pregnant women with singleton gestations, between 11 + 3 and 14 + 0 weeks of gestation, and with crown-rump length (CRL) ≤ 84 mm at the time of randomization; nuchal translucency (NT) ≤ 3.5 mm; and absence of fetal abnormalities at the first-trimester ultrasound screening. Women with multiple gestations, including vanishing twins, and those who have already planned for invasive prenatal testing were excluded. Gestational Age (GA) was calculated according to the last menstrual period and confirmed by fetal CRL measurement for spontaneous pregnancies, and from the day of embryo transfer in case of assisted reproductive techniques (ARTs).

For the purpose of this sub-study, patients with high risk at the first-trimester screening were excluded.

### 2.2. Questionnaires

To measure reassurance, satisfaction, and anxiety, we used specific questionnaires, as proposed in previous studies [14,15].

Questionnaires have been filled in, first, at time of randomization (Q1), and, subsequently (Q2), after receiving screening results.

Maternal, pregnancy-related, and child-related anxiety, satisfaction, and reassurance have been tested.

An Italian version of the six-item short form of the state scale of the Spielberger State–Trait Anxiety Inventory (STAI), described by Marteau et al. [18], has been used in order to measure maternal anxiety, scoring women on a scale of 20–80, where higher scores mean higher levels of anxiety. A STAI score of 34–36 was considered normal anxiety; a STAI score between 37 and 55 was considered mild anxiety.

Pregnancy-Related Anxiety Questionnaire—Revised Regardless of Parity (PRAQ-R2 scale) measured child-related anxiety, using a score from 1 to 5 for each of the 11 items [19,20]. Women were scored on a scale of 11–55, where higher scores mean higher levels of child-related anxiety.

Considering only four items (Item 4, 9, 10, and 11) of PRAQ-R2 permitted us to assess fear of bearing a physically or mentally handicapped child, with a total score ranging from 4 to 20 [20].

In terms of reassurance, in Q1, patients were asked if they were happy with the assigned screening test (FTCS/cfDNA), with a score ranging from 1 (disagree strongly) to 5 (agree strongly). Otherwise, in Q2, reassurance has been evaluated by assigning a score ranging between 5 points to 35 points to three different questions about reassurance after receiving test result, about being confident that test result is correct and about being sufficiently certain whether there is an absence of any fetal disorder.

Satisfaction has been evaluated only during Q2, asking patients if they were happy with the assigned screening test, with a score ranging between 5 points to 35 points. An English version of STAI and PRAQ questionnaires has been reported in Appendix A.

### 2.3. Statistical Analysis

Data are shown as means ± standard deviation (SD), or as absolute number (percentage). Univariate analysis of categorical variables was performed using the chi-square test or Pearson’s exact test. Comparisons between groups were performed with the use of the *t*-test or Mann–Whitney test, as appropriate. Statistical analysis was performed using Statistical Package for Social Sciences (SPSS) v. 29.0.1.0 (IBM Inc., Armonk, NY, USA).

## 3. Results

From 1 November 2022 to 14 April 2024, 620 patients were assessed for eligibility Among them, 434 (70%) eligible patients consented to participate in the study; 3 (0.7%) were excluded after Q1, due to high risk after first-trimester screening (Figure 1).

The characteristics of the remaining 431 patients have been summarized in Table 1. Both groups were comparable for the main anamnestic and demographic characteristics.

All the results of the questionnaires at Q1 and Q2 and the delta value (the mean difference between Q1 and Q2) have been reported in Table 2.

At the time of the randomization (Q1), patients undergoing contingent screening reported a significantly lower reassurance than those undergoing cfDNA (4.94 ± 0.27 vs. 4.99 ± 0.12; *p* = 0.01). However, at Q2, no significant differences in terms of reassurance, satisfaction, and anxiety have been found between groups.

## 4. Discussion

### 4.1. Main Findings

In this randomized trial, we aimed to understand the psychological effects on mothers of two different screening strategies: the first based on a contingent approach, and the second one offering universal cfDNA, after a normal first-trimester screening ultrasound. According to the STAI results, in both groups, patients reported a level of anxiety above the normal, corresponding to a level of mild anxiety.

The trial showed that patients randomized for the contingent screening were less happy before performing the screening, but both strategies were associated with similar maternal reassurance, satisfaction, and anxiety, after receiving a low-risk result.

### 4.2. Comparison with Previous Studies

The mild level of anxiety observed in the study population is consistent with previous studies reporting that rates of anxiety are significantly higher in this maternal population than in the general one [21,22].

Regarding the couple’s experience, this is the first study comparing a contingent model with the universal use of cfDNA, after a normal first-trimester ultrasound scan. Previous studies assessed the psychological impact of different screening strategies. Chueh et al. reported that women with increased NT did not have a sustained increase in anxiety and remained supportive about the value of screening, demonstrating the small psychological impact of a prenatal screening test, even in the case of abnormal results [23]. Similarly, it has been reported that screen-positive results after a detailed ultrasound examination in the first trimester can increase maternal anxiety only temporarily [24].

V van Schendel et al. assessed maternal experience in women performing cfDNA testing after a high risk was detected at the combined screening test; they concluded that cfDNA might be an option satisfying and reassuring even high-risk pregnant women [15].

A similar randomized trial compared women’s experience after the combined screening test, or after an approach using a combination of ultrasound examination and cfDNA analysis. The analysis reported that women randomized to the cfDNA group had a higher satisfaction and lower mean anxiety score, as assessed in the STAI pre-test questionnaire [23]. However, this trial included only 40 women and patients could not have access to cfDNA testing as a second-line screening, in the case of intermediate risk being detected after the combined screening test.

### 4.3. Clinical and Research Implications

A recent systematic review and meta-analysis [25] estimated the prevalence of antenatal and postnatal anxiety, reporting a prevalence rate for self-report anxiety symptoms in the first trimester of 18.2%, which reaches 24.6% in the third trimester; postnatally, the prevalence of maternal experience of anxiety symptoms is 17.8% in the first 4 weeks following childbirth.

Even if perinatal maternal anxiety appears to be a frequent morbidity [26], it has received limited attention from researchers and health professionals. This is an important clinical omission; in fact, it has been proven that maternal anxiety is a leading cause of serious negative outcomes, such as increased childbirth fear, maternal preference for caesarean section, decreased effective coping strategies, higher rates of eating disorders, increased preterm birth rates, and lower Apgar scores. Therefore, we think that it is important to consider the psychological impact of a screening strategy before promoting its implementation. We compared universal cfDNA testing with contingent screening because, even if the latter seems to be more cost-effective in different settings [4,5,6,7,8,9], the use of cfDNA seems to be a very common option in Italy, despite the high costs [3]

Our results reasonably encourage the use of the contingent model as first-trimester screening, which not only can be considered an effective option in terms of specificity and sensitivity [1,4,27], but also is adequately accepted by patients. Indeed, those women undergoing contingent screening reported similar scores of reassurance, anxiety, and satisfaction as a cfDNA analysis after a low-risk result.

### 4.4. Strengths and Limitations

The main strength of this study is the large sample size. Moreover, all the included patients were counseled by the same medical team, which highlighted, always in the same way, the strengths and limitations of both screening protocols to the patients. Finally, the questionnaires were not self-administrated, but patients received constant support from a specialized psychologist.

To the best of our knowledge, this is the first study comparing psychological impact in these two screening protocols. However, we did not report data on the psychological impact of the protocols in the high-risk group, due to the small number of patients in this group. A further study will also address this aspect. Moreover, even if the two groups were similar for the main maternal characteristics, we cannot exclude possible effects of hidden confounders on the results, such as a previous history of stillbirth.

## 5. Conclusions

According to our results, the implementation of contingent screening for aneuploidies in the first trimester seems reasonable in terms of maternal psychological impact, even where the use of a cfDNA analysis is largely spread in private settings. Indeed, it seems able to ensure the same maternal reassurance and satisfaction as cfDNA analysis in the low-risk population and to not affect maternal anxiety.

## Figures and Tables

**Figure 1 diagnostics-14-01198-f001:**
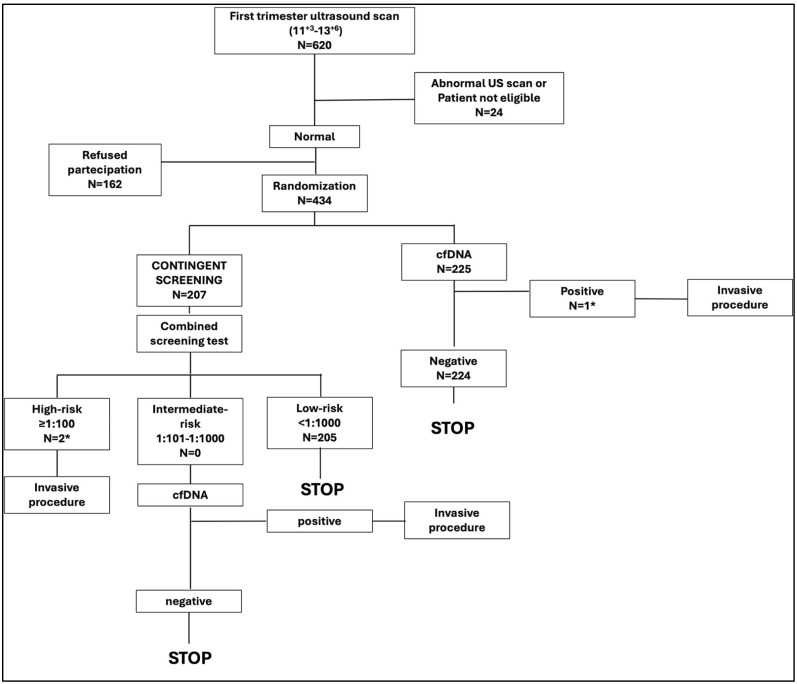
Study protocol. * Patients excluded after Q1, due to high risk at first-trimester screening.

**Table 1 diagnostics-14-01198-t001:** Maternal characteristics in the two study groups.

Maternal Characteristics	Contingent Screening (*n* = 205)	Universal cfDNA (*n* = 226)	*p* Value
Maternal age	32.10 ± 6.35	32.20 ± 24.96	0.720
Caucasian ethnicity	200 (97.6)	223 (98.7)	0.486
Smoking	28 (13.7)	21 (9.3)	0.173
Nulliparity	102 (49.8)	124 (54.9)	0.483
Previous preeclampsia	12 (5.8)	11(4.8)	0.674
ART conception	19 (9.3)	19 (8.4)	0.865
Pre-existing chronic condition	30 (14.6)	25 (11.1)	0.227
**Education:**			0.279
Primary school or less	59 (28.8)	50 (22.2)	
Secondary school	64 (31.2)	79 (34.9)	
Degree	74 (36.1)	82 (36.3)	
Unknown	8 (3.9)	15 (6.6)	
**Occupation:**			0.371
Unemployed	74 (36.1)	73 (32.3)	

**Table 2 diagnostics-14-01198-t002:** Results of the questionnaires about reassurance, satisfaction, and anxiety in the two groups.

Results	Contingent Screening (*n* = 205)	Universal cfDNA (*n* = 226)	*p* Value
Mean Reassurance Q1	4.94 ± 0.264	4.99 ± 0.115	**0.010**
Mean Reassurance Q2	74.12 ± 22.87	73.38 ± 20.93	0.725
Mean STAI Q1	50.37 ± 6.96	50.56 ± 7.55	0.792
Mean STAI Q2	52.47 ± 7.49	52.66 ± 7.59	0.797
Delta STAI (Q1-Q2)	−2.10 ± 9.20	−2.10 ± 10.19	1.00
PRAQ Q1	23.53 ± 8.63	24.51 ± 8.81	0.244
PRAQ Q2	25 ± 9.28	26.2 ± 9.29	0.18
Delta PRAQ (Q1-Q2)	−1.47 ± 8.33	1.69 ± 8.54	0.785
Mean Subscale 4-item PRAQ-R2 Q1	9.63 ± 5.18	10.28 ± 5.05	0.188
Mean Subscale 4-item PRAQ-R2 Q2	8.72 ± 4.88	9.58 ± 4.72	0.063
Delta Subscale 4-item PRAQ-R2 (Q1-Q2)	0.92 ± 4.77	0.73 ± 4.74	0.682
Mean Satisfaction Q2	26.29 ± 8.74	28.02 ± 7.79	0.051

## Data Availability

The original contributions presented in the study are included in the article; further inquiries can be directed to the corresponding author.

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
