# Peer review of "Maternal Reassurance, Satisfaction, and Anxiety after First-Trimester Screening for Aneuploidies: Comparison between Contingent Screening and Universal Cell-Free DNA Testing"

_diagnostics, 2024, doi:10.3390/diagnostics14111198_

Round 1

Reviewer 1 Report

Comments and Suggestions for Authors

This is an interesting paper dealing with an important problem – an impact of different screening strategies for aneuploidies on mother’s psychological state. The experimental setting appears to be correct. However, a few points should be addressed before considering the paper for publication

1.      I would recommend to add a scheme of the study protocol instead of text (pages 81-93) since the scheme is more convenient and clear for the readers. The number of participants in each group should be also included.

2.      Adding at least a few items from the questionnaires would be helpful.

3.      Table 2 and lines 151-152: I don’t believe that the values of Mean Reassurance Q1 of 4.94+0.264 (Contingent screening) and 4.99+0.115 (Universal cfDNA) are statistically different (5/100), especially taking into account the SD values are overlapping and much greater. Please, check and explain if it is true.

Comments on the Quality of English Language

Minor English editing is required.

Author Response

  1. I would recommend to add a scheme of the study protocol instead of text (pages 81-93) since the scheme is more convenient and clear for the readers. The number of participants in each group should be also included.

We thank the reviewer for the kind suggestion. We included a scheme to summarize the study protocol reporting also the number of participants in each group (see Figure 1). We did not amend the text because we thought that an explanation of the study protocol also in methods could make it clearer for the readers.

  1. Adding at least a few items from the questionnaires would be helpful.

We thank the reviewer for the kind suggestion.We added an English version of the questionnaires as Supplementary Material.

3.    Table 2 and lines 151-152: I don’t believe that the values of Mean Reassurance Q1 of 4.94+0.264 (Contingent screening) and 4.99+0.115 (Universal cfDNA) are statistically different (5/100), especially taking into account the SD values are overlapping and much greater. Please, check and explain if it is true.

Thank you for this observation. However, we checked again and we confirm this result.

Reviewer 2 Report

Comments and Suggestions for Authors

'Maternal reassurance, satisfaction, and anxiety after first trimester screening for aneuploidies: comparison between contingent screening and universal cell-free DNA testing' is an original article with the aim to investigate maternal reassurance, satisfaction and anxiety after two different noninvasive prenatal strategies performed in the first trimester of pregnancy (contingent screening vs. cfDNA testing). Results of the study showed that patients randomized for the contingent screening were less happy before performing the screening, but both strategies were associated with similar maternal reassurance, satisfaction, and anxiety, after receiving a low- risk result.

Manuscript is written in a well-structured manner. Quality of English used is fine. All the cited references are relevant for the field and relatively new. Ethics statement is adequate. Results are presented in two tables. Data presented is easy to understand. Conclusions are consistent with the evidence and arguments presented. Authors have stated limitations and strengths of their study.

Final message of this study is reasonable encouragement to recommend first trimester contingent screening to all the pregnant women since the level of the anxiety after the perfomed test is similar to the cfDNA testing and its' low cost.

There are some minor typing errors to be corrected (for example in lines 109 and 110).

Therefore, I recommend this article to be accepted after minor revisions.

Author Response

There are some minor typing errors to be corrected (for example in lines 109 and 110).

We thank the reviewer for his/her positive comments.

We corrected typing errors.